# Time from Symptom Onset to Diagnosis and Treatment among Haematological Malignancies: Influencing Factors and Associated Negative Outcomes

**DOI:** 10.3390/medicina55060238

**Published:** 2019-06-03

**Authors:** Austėja Dapkevičiūtė, Virginijus Šapoka, Elena Martynova, Valdas Pečeliūnas

**Affiliations:** 1Haematology, Oncology and Transfusion Medicine Centre, Vilnius University Hospital Santaros Klinikos, Santariškių 2, LT-08661 Vilnius, Lithuania; valdas.peceliunas@santa.lt; 2Institute of Clinical Medicine, Vilnius University, Santariškių 2, LT-08661 Vilnius, Lithuania; virginijus.sapoka@santa.lt; 3Quantitative Psychology Program, University of Virginia, 485 McCormick Road Charlottesville, VA 22903, USA; em6gg@virginia.edu

**Keywords:** diagnostic lagtime, diagnostic delay, multiple myeloma, Hodgkin lymphoma, Non-Hodgkin lymphoma

## Abstract

*Background and objectives*: Diagnostic delay causes unfavorable outcomes among cancer patients. It has been widely analyzed in solid tumors. However, data regarding hematological malignancies diagnostic delay are scarce. We aimed to evaluate diagnostic intervals, their influencing factors, and the negative effect on clinical outcomes among multiple myeloma and lymphoma patients. *Materials and methods*: One hundred patients diagnosed with multiple myeloma (*n* = 53) or lymphoma (*n* = 47) (ICD codes—C90, C81–C84) were asked to participate during their scheduled hematology consultations. Interval durations and the majority of influencing factors were assessed based on a face-to-face questionnaire. Data of disease characteristics were collected from medical records. *Results:* The median interval from symptom onset to registration for medical consultation was 30 (0–730) days, from registration to consultation 2 (0–30) days, from first consultation to diagnosis 73 (6–1779) days, and from diagnosis to treatment 5 (0–97) days. Overall time to diagnosis median was 151 (23–1800) days. Factors significantly prolonging diagnostic intervals in multivariate linear regression were living in big cities (*p* = 0.008), anxiety and depression (*p* = 0.002), self-medication (*p* = 0.019), and more specialists seen before diagnosis (*p* = 0.022). Longer diagnostic intervals resulted in higher incidences of multiple myeloma complications (*p* = 0.024) and more advanced Durie-Salmon stage (*p* = 0.049), but not ISS stage and Ann-Arbor staging systems for lymphomas. *Conclusion:* Median overall diagnostic delay was nearly 5 months, indicating that there is room for improvement. The most important factors causing delays were living in big cities, anxiety and depression, self-medication, and more specialists seen before diagnosis. Diagnostic delay may have a negative influence on clinical outcomes for multiple myeloma patients.

## 1. Introduction

Multiple myeloma (MM), Hodgkin lymphoma (HL), non-Hodgkin lymphoma (NHL) are the most common hematological malignancies [1,2]. According to the Lithuanian Hematology Monitoring System database, there are 100 new cases of HL, 400 of NHL, and around 160 new cases of MM every year in Lithuania.

Diagnostic delay is usually defined as the time from symptom onset until correct diagnosis [3]. There are a lot of data about diagnostic delay and associated negative outcomes [4,5,6,7]. However, the majority of these studies have focused on solid tumors [4,5,6,7,8,9]. Data gathered from studies evaluating solid tumors cannot be applied to hematological malignancies due to their rarity, insidious clinical presentation, and the need for specific laboratory tests resulting in difficult diagnostic pathways usually requiring multiple primary care consultations [10,11,12].

Few studies have focused specifically on diagnostic delay of hematological malignancies. However, different authors have provided varying results regarding time to diagnosis duration, its influencing factors, and associated adverse outcomes [2,11,13,14,15,16,17,18,19]. Variations might occur because of regional differences in healthcare systems, education, psychological, and social factors. On the other hand, the majority of the previously mentioned studies [2,11,13,14,15,16,17] were based on retrospective medical record analyses. This might lead to mismatches due to differing precision in documenting the pathway from symptom onset to diagnosis [3,18]. To the best of our knowledge, there are only two studies in which adult patients were interviewed (one of them via a postal questionnaire) [11,18]. They included a high number of patients, but focused more on establishing time-to-diagnosis and symptoms of hematological malignancies than on analyzing influencing factors and adverse outcomes of diagnostic lag time. Therefore, we aimed to provide more accurate results using a face-to-face questionnaire in addition to medical documentation analysis.

We aimed to evaluate diagnostic delay, disease, patient and health-system-related influencing factors and the effects of longer times to diagnosis among multiple myeloma and lymphoma patients.

## 2. Materials and Methods

### 2.1. Patient Selection

Patients treated at a single institution from January 2017 to April 2017 were included in the study if they met inclusion criteria: diagnosed with MM according to International Myeloma Working Group (IMWG) criteria [20] (ICD code—C90), HL (ICD code—C81) or NHL (ICD code—C82. C83. C84), were ≥18 years, and signed Informed consent form. Patients were excluded if they were unable to adequately answer questions. The study was approved by Vilnius Regional Biomedical Research Ethics Committee (No. 158200-17-895-404) on 10 January 2017.

### 2.2. Study Design

Patients were invited to participate in the study during their scheduled hematology consultations in the outpatient clinic, irrespective of their disease and treatment status. The majority of data were gathered from our own face-to-face questionnaire based on former studies [2,3,9,13,14,21,22,23]. Disease-related information (stage, complications, diagnosis date, etc.) and basic demographic information (age, place of residence, etc.) were collected from medical records. The exact questionnaire can be found in Appendix A: Sociodemographic data collected from questioning participants, Appendix A: Questions about symptom evaluation, reasons for delay and other patient-related factors, Appendix A: Questions about health system related factors, Appendix A: Data gathered from medical records. The study design is overviewed in Figure 1.

We evaluated the duration of time to diagnosis and treatment divided into several intervals based on systematic review [3]: A interval—time between symptom onset and registration for first medical consultation, B interval—time between registration for first medical consultation and first medical consultation, C interval—time between first medical consultation and diagnosis, D interval—time between diagnosis and treatment, E interval—overall time between symptom onset and diagnosis (sum of A, B, C intervals), F interval—overall time between symptom onset and treatment (sum of A, B, C, D intervals) (Figure 1). Disease, patient and health-system-related factors which might influence time to diagnosis were collected (analyzed factors are shown in Figure 1 and Table 1). We also evaluated the possible negative effect of diagnostic lag time on disease stage (Durie-Salmon and ISS stage for MM, Ann-Arbor stage for lymphoma) and burden of complications at the time of diagnosis.

### 2.3. Statistical Analysis

Assumptions of normality were tested using Kolmogorov-Smirnov and Shapiro-Wilk tests. Patient characteristics, disease-related factors, and health-system-related factors were summarized using descriptive statistics. The categorical influence of variables on interval duration was assessed using Kaplan-Meier curves and Cox regression analysis. The linear effects of variables on interval duration were measured using univariate linear regression. Results were considered to be significant at a 5% level of significance. Multivariate analysis with dependent variables as 5 different time intervals was conducted using linear regression. Interval duration distribution was improved with log transformation. Therefore, we provided original Beta values from log-transformed data as well as percentile effect of independent variables each one-unit increase on untransformed dependent variable (percentile Beta values). The latter was calculated as e^Beta value^ − 1. Categorical and linear patient, disease and health-system-related factors were chosen as independent variables. In order to get the best models, different selection algorithms were used: Mallow’s Cp, Akaike Information Criterion (AIC) (forward AIC, backward AIC and both AIC), Schwarz’s Bayesian Information Criterion (BIC), adjusted R-squared. The best models were selected based on Leave-one-out cross-validation, residual standard error, R squared, and F statistic values. A *p* value equal to 0.15 cutoff for determining factor inclusion and removal was used. To assess the negative impact of longer times until diagnosis (E interval), we used multivariate logistics and Poisson regression analysis. Statistical analysis was carried out using IBM Statistical Package for the Social Sciences (version 22.0 for Windows; SPSS Inc, Chicago, IL, USA) and RStudio Inc.

## 3. Results

### 3.1. Patient Characteristics

We included 100 patients: 53 (53.0%) were diagnosed with MM, 47 (47.0%) with lymphomas—21 (44.7%) HL and 26 (55.3%) NHL.

The majority of patients diagnosed with lymphomas had stage IV disease—26 (55.3%) (Appendix A: Lymphomas distribution according to disease stage). The most common NHL type was diffuse large B cell lymphoma—10 (38.5%) (Appendix A: Lymphomas distribution according to NHL type). Overall, 9 (34.6%) NHL patients had indolent disease, and 17 (65.4%) had aggressive disease. MM patient distribution according to disease stage was: 21 (39.6%), 14 (26.4%), 18 (34.0%) for I, II, III ISS stages and 9 (17.0%), 14 (26.4%), 30 (56.6%) for I, II, III Durie-Salmon stages respectively.

Forty (75.5%) patients had MM associated complications at the time of diagnosis. Established MM related complications were bone lesions 31 (58.5%), anemia 16 (30.2%), renal insufficiency 10 (18.9%), infectious syndrome (measured as 2 or more bacterial infections in 12 months) 9 (17.0%), and hypercalcemia 1 (1.9%). Eighteen (34.0%) patients had 2 or more complications at the time of diagnosis.

For all patients, the median number of disease-related symptoms was 2 (0–5). B symptoms, defined as unexplained weight loss of >10% of body weight in the 6 months preceding the diagnosis, unexplained fever with temperatures >38 °C, and drenching night sweats, were reported by 27 (27.0%) participants. Summary of patient and health-system-related characteristics are shown in Table 1. Also, participants were asked about the most important reasons causing delay in A, B, C, D intervals. A summary of their responses is shown in Appendix A: The most important reasons for delay in A, B, C, D intervals according to participants. In addition, diagnostic pathways are demonstrated in Appendix A: MM and lymphomas diagnostic pathways.

### 3.2. A, B, C, D, E, F Interval Durations

For all patients, the A interval median was 30 days (0–730), B interval 2 (0–30), C interval 73 (6–1779), D interval 5 (0–97), E interval 151 (23–1800), and F interval 157 days (24–1803).

When comparing different time intervals among the same individual using non-parametric related samples tests, we found statistically significant differences among A, B, C, D interval durations (*p* < 0.001). After post-hoc analysis with Bonferroni correction, C interval was found to be the longest, followed by A interval duration (*p* < 0.001 in all cases). B and D intervals were the shortest, and did not differ statistically significantly from each other (*p* = 0.056).

### 3.3. Disease-Related Factors’ Influence on Diagnostic Lag Time

Intervals of MM, HL, NHL are shown in Figure 2. Only the D interval was statistically significantly shorter for MM patients in comparison with both—HL and NHL (*p* < 0.001 in both cases). Other interval durations were not statistically significantly different for MM, HL, NHL patients.

Patients with B symptoms tended to have shorter E (HR = 0.66, 95% CI 0.421–1.034, *p* = 0.066) (Figure 3a) and F (HR = 0.67, 95% CI 0.431–1.058, *p* = 0.083) intervals. Participants with more symptoms had longer B interval (*p* = 0.022) duration.

### 3.4. Influence of Patient-Related Factors on Diagnostic Intervals

The effects of patient-related demographic parameters (Table 1) on A, B, C, D, E, F interval durations were analyzed. We found an association between self-medication and prolonged A (HR = 1.53, 95% CI 0.971–2.421, *p* = 0.061), E (HR = 1.56, 95% CI 0.993–2.453, *p* = 0.051) (Figure 3b), and F (HR = 1.54, 95% CI 0.982–2.428, *p* = 0.0571) intervals. Patients with higher HADS score had significantly longer C (*p* = 0.001), E (*p* = 0.001) and F interval duration (*p* = 0.001). Higher CIRS (*p* = 0.020) score was observed when B interval was shorter.

### 3.5. Health-System-Related Factors’ Influence on Interval Durations

Patients from the largest cities in Lithuania (Vilnius n = 51, Kaunas n = 4) had longer time from symptom onset to diagnosis (E interval) (HR = 1.65, 95% CI 1.100–2.488, *p* = 0.016) (Figure 3c) and from symptom onset to treatment (F interval) (HR = 1.65, 95% CI 1.098–2.483, *p* = 0.016). A tendency that living in Vilnius/Kaunas prolonged B (HR = 1.39, 95% CI 0.930–2.089, *p* = 0.068) and C intervals (HR = 1.39, 95% CI 0.928–2.083, *p* = 0.106) was also observed.

We found that C (HR = 3.17, 95% CI, *p* < 0.001), E (HR = 1.61, 95% CI 1.040–2.497 *p* = 0.033) and F (HR = 1.62 95% CI 1.045–2.509, *p* = 0.032) intervals were longer when an oncologic diagnosis was not suspected after a first visit (Figure 3d). Also, a clinician prescribing treatment before correct diagnosis resulted in longer C interval (HR = 1.20, 95% CI 1.289–2.968, *p* = 0.001) (Figure 3e). In addition, when the second consulted doctor was not a hematologist, the C interval was longer (HR = 2.58, 95% CI 1.382–4.828, *p* < 0.001) (Figure 3f).

When a participant visited more specialists before a correct diagnosis was made, the C (*p* < 0.001), E (*p* = 0.002), and F (*p* = 0.002) intervals were prolonged.

### 3.6. Influence of Disease, Patient and Health-System-Related Factors on Interval Durations—Multiple Linear Regression

We evaluated disease-related (diagnosis—MM or lymphomas, B symptoms, number of symptoms), patient-related (Table 1), and health-system-related (except for A, B interval durations) (Table 1) factors effects on A, B, C, D, E intervals. None of the factors substantially predicted A interval duration. To predict other interval durations—statistically significant multiple linear regression models were calculated and are shown in Table 2.

### 3.7. Effect of Diagnostic Delay on Disease Stage and Complications at the Time of Diagnosis

Multiple logistic and Poisson regression models with dependent factors as MM complications, MM Durie-Salmon stage, ISS stage for MM and Ann-Arbor stage for lymphomas were calculated. Independent factors were chosen as E interval duration, age, CIRS score, and gender. E interval duration was shown to be an independent risk factor associated with increased burden of complications and increased MM Durie-Salmon stage (models shown in Table 3). Notably, renal insufficiency and anemia (tendency) were more common for patients with longer E interval duration when corrected for the same independent variables (*p* = 0.037 and *p* = 0.092 respectively). MM ISS stage was not significantly influenced by E interval duration. Also, in lymphoma patients, Ann-Arbor stage was not statistically significantly associated with a longer E interval duration.

## 4. Discussion

### 4.1. Interval Durations and Comparison with Other Studies

In this study, we analyzed a patient pathway from symptom onset to treatment initiation. We also investigated negative consequences of prolonged time until diagnosis.

For lymphoma patients, time from symptom onset to a first medical consultation (A + B interval duration 17 days for HL, 25 for NHL) was similar to data reported from studies performed in the UK and Canada [11,14]. These time intervals are comparable to non-hematological malignancies, varying approximately from 14 to 30 days [9,24].

On the other hand, for MM patients we found twice as long a time from symptom onset to first medical consultation (56 days) in comparison with our lymphoma patients and MM patients described in other studies [9,11,24]. Our questionnaire revealed that MM patients more often interpret initial symptoms as non-significant (e.g., think they will resolve on their own—Appendix A: the most important reasons for delay in A, B, C, D intervals according to participants) than lymphoma patients (58.5% and 46.8% respectively). Such underestimation of the initial symptoms may have led to a higher rates of self-medication (34.0% for MM, 19.1% for lymphoma patients) which proved to be an independent risk factor predicting longer overall times until diagnosis.

Interval B in our study was very short and did not significantly contribute to the overall time from symptom onset to diagnosis (B interval median 2, range 0-30 days). There is lack of data regarding B interval duration, since the majority of studies attributed time from registration until first consultation to “patient-related delay”, and measured overall time from symptom onset to first medical consultation [9,11,14,17,24,25,26,27].

C interval, from first medical consultation to diagnosis, was the longest (median 73 days), and was similar across all analyzed diseases. Researchers in the UK found approximately 1–2 week longer C interval equivalent for lymphoma and MM patients in comparison with our results [11]. It is important to note that in our study, the vast majority of included patients (roughly around 90%) addressed acquaintances working in healthcare system in order to reach secondary medical specialists more quickly, and expressed the idea that reaching needed specialists otherwise would have taken much more time. We did not intend to collect this information, but we suggest that this might be an important factor which should be addressed in future research.

D interval duration—2 days for MM, 8 for HL, 7 for NHL was similar to data from Canada—5 days for NHL [14] and considerably shorter in comparison with UK—17.5 days for all lymphomas [17] or India—59 days for all cancer types [9]. B interval together with D interval heavily depends on healthcare system, particularly waiting lists. These two intervals summed up had a median of 9 days (range 0—99 days) and accounted for only 5.7% of all overall duration from symptom onset to treatment initiation. The sum of the maximum values (54, 99 days) of B and D intervals were due to patient indecisiveness to start treatment and not due to long waiting lists. These study results show that healthcare systems, particularly waiting lists, from registration to first consultation and from diagnosis to treatment, work adequately with little room for improvement.

Our established overall intervals to diagnosis (E interval—175 days for MM, 174 for HL, 132 days for NHL) and treatment (F interval—177 for MM, 181 for HL, 137 for NHL patients) are similar to those reported from the United Kingdom, Canada, Germany or Hungary [11,14,17,18,24,28]. However, we are falling behind in comparison with the US – E interval analog was only 99 days for MM [13], 63 days for HL, and 34 days for NHL [16]. It is unclear what causes these differences, since the studies from the US did not provide separate intervals, only overall time to diagnosis. The latter study analyzed only young adults (15–29 years) and evaluated time until pathological confirmation [16], while we evaluated time until full clinical assessment (including PET-CT scans when needed); this could explain some of the differences. The majority of cited studies [13,14,16,17,24,28] were based on retrospective medical documentation analysis which might lead to inaccuracies accounting for up to 2 months as a recent study revealed [18]. Nevertheless, the gap between our results and those obtained from the US indicates that the overall time to diagnosis can be considerably shorter. Since our B and D interval durations were acceptable already, in order to decrease overall diagnostic delay, attention should be focused on shortening time from symptom onset to registration for a medical consultation (A interval) and from the first consultation to diagnosis (C interval).

### 4.2. Disease-Related Factors’ Effect on Diagnostic Lag Time

An E and F interval shortening tendency was observed in the presence of B symptoms. Similar results stating that B symptoms shorten diagnostic intervals were provided by other researchers [14,29].

Also, we found that once diagnosed, MM treatment is started sooner than lymphoma treatment (*p* < 0.001). This observation might be due to the fact that in our hospital MM treatment is usually started at the Day clinic, while HL and NHL treatment mostly begins at the inpatient department. This takes more time due to longer waitlists for hospitalization.

### 4.3. Patient-Related Factors’ Effect on Interval Durations

Patients with higher HAD scale value had significantly longer C, E, F interval durations. We do not have a good explanation for this finding, but in some cases, GPs (general practitioners) might misinterpret symptoms caused by hematological malignancy as being anxiety or depression related. This suggestion is based on data from existing literature regarding solid tumors, saying that GPs are more likely to pay attention to abnormal symptoms (such as abdominal pain or bloating for colon cancer or breast lumps for breast cancer) than abnormal laboratory results (such as iron deficiency anemia), even though sometimes abnormal laboratory results are much better cancer predictors [30]. For patients with anxiety, GPs might disregard new symptoms as another anxiety-related episodes, and for patients with depression, fatigue or one of B symptoms, weight loss can be viewed as being depression-related.

The higher CIRS comorbidity index for our patients shortened B interval duration and did not influence the other intervals. We can speculate that patients with more comorbidities may have a higher awareness of new worrying symptoms, know their primary care physicians better, and visit them more frequently due to other comorbidities. We could not confirm these findings from other researchers [13,18] stating that a higher number of comorbidities resulted in longer diagnostic delays, possibly due to our healthcare system peculiarities.

Self-treatment was a significant factor prolonging A, E, F interval durations. None of the patients’ demographical characteristics shown in Table 1 significantly discriminated those who self-medicated and those who did not, although a non-significantly higher percent of self-treatment was found among patients with higher education (31.5% vs. 21.7%, *p* = 0.274). According to other studies, self-medication is either more common among patients with lower medication knowledge and lower self-efficacy [31], or younger patients with higher levels of education and socioeconomic statuses, or those who have to wait longer for medical care consultations [32,33,34]. MM patients tended to self-medicate more than lymphoma patients (34.0% vs. 19.1%, *p* = 0.117). Those patients who named service barriers (“I would be worried about wasting the doctor’s time”, etc. [35]) as one of the reasons for patient-related delay self-medicated significantly more often (71.4% vs. 23.7%, *p* = 0.006). This result should be evaluated with caution since only 7 patients named service barriers as one of the reasons for delay and none of them named it as the most important one (Appendix A: The most important reasons for delay in A, B, C, D intervals according to participants). Oncologic diagnosis after a first consultation was suspected more often for those patients who did not self-medicate (37.0% vs. 11.1%, *p* = 0.012). Primary care physicians tended to prescribe treatment without a clear diagnosis for the self-medication group more frequently (63.0% vs. 54.2%, *p* = 0.115). From these results, it seems that if a patient decides to medicate him/herself without knowing what caused his symptoms, primary care physicians are also less alert to suspect cancer and are more likely to treat patients symptomatically before establishing a diagnosis.

Our results highlight the importance of patient ability to understand their symptoms and react accordingly. There are examples from other countries with nationwide campaigns directed at informing patients about the most common cancer signs, and advising them to seek professional advice without delay [36].

### 4.4. Health-System-Related Factors’ Effect on Diagnostic Lag Time

Oncologic diagnosis not suspected at the first visit, as well as a higher number of specialists before correct diagnosis, statistically significantly increased C, E, F interval durations. In addition, prescribed treatment before correct diagnosis resulted in a longer C interval. Other studies concluded that a higher number of specialists seen before correct diagnosis [9,37], as well as incorrect preliminary diagnosis and prescribed symptomatic treatment, elongates C interval equivalent [9]. These results are in line with ours.

It has been noted [21,22] that GPs rarely feel confident informing patients about hematological diseases, and are not well informed about diagnostic tests for MM, and avoid using them. More common use of these tests could help to reduce diagnostic delay [38,39]. In Lithuania, serum or urine protein electrophoresis is not reimbursed in a primary care setting. Consequently, the first step should be making these tests available followed by guidelines helping to choose patients for screening and evaluate results obtained. An example could be initiatives like HOPING (Haematologic Oncology Primary Intervention Networking Group), developed to educate primary care physicians on the signs and symptoms of haematologic malignancies and to encourage timely referrals [21,22]. Nationwide projects with reliable decision-making aids for physicians directed specifically at reducing the longest physician-related interval delay should be a priority of our healthcare system after making the main screening tests widely available.

Our finding that living in the largest cities in Lithuania—Vilnius (n = 51) or Kaunas (n = 4)—is associated with longer B (tendency), C, E, and F interval durations is somewhat controversial. Researchers from India [9] and Germany [18] reached different conclusions—living in rural areas had a negative impact on A + B interval equivalent mainly due to poor access to GPs and economic problems [9,37]. None of our patients named financial issues or distance to the primary care specialist as the main reason for a patient-related delay. The main barriers preventing seeking help were symptom misinterpretation or practical reasons (such as being busy) (Appendix A: The most important reasons for delay in A, B, C, D intervals according to participants). Lithuania is 9th in the world according to density of physicians (total number per 1000 population), with better access to medical personnel in comparison with India or Germany [40]. Also, patient number for 1 GP in the largest cities is similar to smaller towns (unpublished data from the National Health Insurance Fund under the Ministry of Health) (Appendix A: Number of patients for 1 PCP (primary care physician—including GPs and general pediatricians), in Lithuanian counties and biggest cities). These data explain the absence of GPs accessibility problems and the longer diagnostic delays in smaller towns which were observed in previous studies [9,18,37]. However, we do not have a good explanation for a longer diagnostic delay in the largest cities; nevertheless, it is an interesting finding which invites further investigation. In addition, the only other study regarding cancer diagnostic delay done in Lithuania provided different results from ours—living in bigger cities shortened breast cancer diagnostic delay through a patient-related interval, which resulted in a significantly shorter overall diagnostic delay [23]. In our study, living in the biggest cities increased the overall diagnostic delay through the physician-related C interval. Such controversies might exist because breast cancer is easier to detect [38], and therefore, patients from bigger cities with higher levels of education and more frequent contact with breast cancer prevention programs might be more aware and visit primary care specialists sooner. In contrast, hematological malignancies are harder to detect due to a non-specific presentation and lack of preventative programs erasing patient-related interval difference between those from rural and urban areas.

### 4.5. Diagnostic Delay Effect on Disease Stage and Burden of Complications

We found that MM patients with longer E interval durations had more complications and higher Durie-Salmon stages at the time of diagnosis (stratified for age, gender, CIRS score). Longer E interval duration did influence ISS stage. It is well known that Durie-Salmon stage mostly represents disease burden while ISS staging depends more on disease biologic activity. Similar findings regarding Durie-Salmon stage have been mentioned before [2,19], while a study done in Hungary also did not find an association between ISS stage and diagnostic delay [28].

Longer times to diagnosis did not have a significant impact on Ann-Arbor lymphoma stage. Published data regarding this subject are controversial—some researchers did not notice a connection between lymphoma stage and a longer time until diagnosis [30], while others found an association with more advanced disease [16,41].

### 4.6. Strengths and Limitations

Only few previous studies analyzed hematological malignancies diagnostic delay based on interviewing patients in addition to medical documentation analysis (one of them via postal questionnaire) [11,18]. Assessing times to diagnosis from medical documentation might only lead to inaccuracies accounting for up to 2 months [18]. This study is one of the few to have analyzed hematological malignancies diagnostic delays based on interviewing patients face-to-face and reviewing their medical records. To the best of our knowledge, we are also the first in the Baltic countries to have addressed hematological malignancies diagnostic delay; previously, only breast cancer time to diagnosis had been assessed in this region [23]. In addition, the connection between diagnostic delay and MM complications, as well as Durie-Salmon stage, was established; however, it was based on only percentile increase, when diagnostic delay was more than 6 months [2] or a higher number of complications when a patient presented via emergency route [19]. We are the first to confirm these findings based on binary logistic regression stratified for age, gender, and CIRS score. Another strength of this study was that the time from symptom onset to diagnosis and treatment was divided into 4 clear time intervals based on systemic review done in 2012 [3]—time from symptom onset to registration for a visit, from registration to first medical consultation, from first consultation to diagnosis, and from diagnosis to treatment. Such an interval division proved that neither waiting time for a first medical consultation, nor waiting time for treatment initiation, significantly enhanced the delay from symptom onset to treatment initiation. While the interval from diagnosis to treatment initiation was noted in few studies [14,17], we were not able to find a single study regarding hematological malignancies that also evaluated waiting time for a first medical consultation.

However, interviewing participants face-to-face has its own limitations. Mostly, such a study design did not allow us to analyze possible future complications and survival differences of prolonged diagnostics and treatment delays. This subject has not been widely assessed, although there is some data to suggest that longer time to diagnosis had a significant effect on disease-free survival [2], but did not have any effect on future complications [9]. Data regarding overall survival differences is conflicting, since a few studies did not find worse overall survival with prolonged diagnostic delay [2,28], while one research found that the emergency presentation route was connected with worse survival for MM patients [19]. This might be an interesting direction for further studies regarding MM diagnostic delay.

## 5. Conclusions

The median overall diagnostic delay was 5 months, indicating that there is room for improvement. The highest impact on overall diagnostic interval was due to patient-related interval A and physician-related interval C, while waitlists for primary care physician and treatment initiation were short and did not have a major influence on diagnostic delays. Living in big cities, higher anxiety and depression levels, self-medication, and more specialists seen before diagnosis significantly increased time until diagnosis. A longer diagnostic delay might have negative clinical consequences. We suggest that nationwide education projects for patients and physicians, as well as making the main diagnostic tests widely available in a primary care setting, could help to reduce diagnostic delays for multiple myeloma and lymphoma patients.

## Figures and Tables

**Figure 1 medicina-55-00238-f001:**
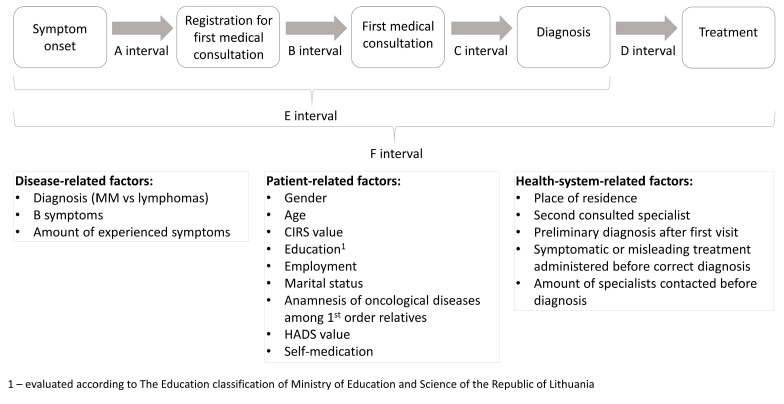
Evaluated diagnostic intervals and factors possibly influencing their duration. CIRS—cumulative illness rating scale; HADS—hospital anxiety and depression scale validated in Lithuania.

**Figure 2 medicina-55-00238-f002:**
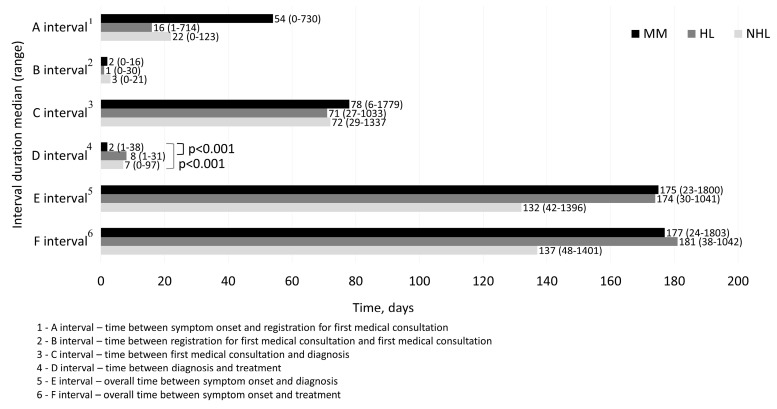
MM, HL, NHL diagnostic interval durations. MM—multiple myeloma; HL—Hodgkin lymphoma; NHL—non-Hodgkin lymphoma.

**Figure 3 medicina-55-00238-f003:**
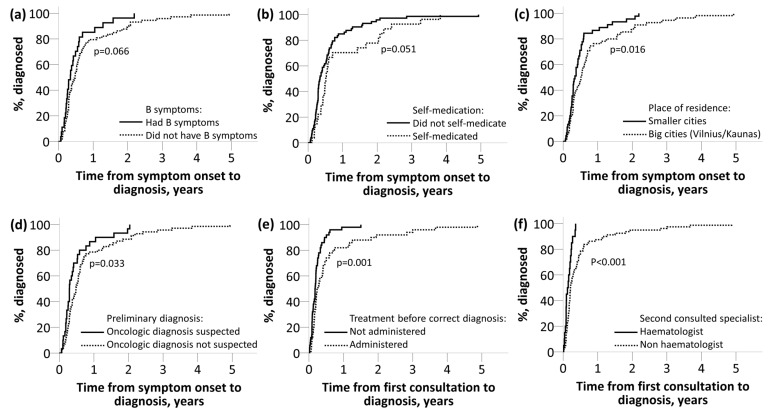
Disease, patient and health-system-related factors influence on time from symptom onset to diagnosis (E interval) or time from first consultation to diagnosis (C interval). (**a**): B symptoms influence on time from symptom onset to diagnosis (E interval); (**b**): Self-medication influence on time from symptom onset to diagnosis (E interval); (**c**): Place of residence influence on time from symptom onset to diagnosis (E interval); (**d**): Preliminary diagnosis influence on time from symptom onset to diagnosis (E interval); (**e**): Treatment before correct diagnosis influence on time from first consultation to diagnosis (C interval); (**f**): Second consulted specialist influence on time from first consultation to diagnosis (C interval).

**Table 1 medicina-55-00238-t001:** Summary of patient and health-system-related characteristics.

Patient-Related Factors	MM	Lymphoma
Gender—male/female N	26/27	21/26
Age (years) median (range)	53 (44–84)	47 (19–81)
CIRS ^1^ median (range)	5 (3–12)	5 (3–11)
Education—no higher education/higher education N	26/27	20/27
Employment before disease—unemployed/employed N	16/37	15/32
Marital status—married/not married N	37/16	24/23
Anamnesis of oncological diseases among 1st order relatives—no/yes N	32/21	34/13
HADS ^2^ median (range)	7 (0–24)	7 (0–23)
Self-medication—no/yes N	35/18	38/9
**Health-system-related factors**		
Place of residence—Vilnius/Kaunas/smaller cities N	27/1/25	24/3/20
First consulted doctor—general practitioner/other specialist N	44/9	42/5
Second consulted doctor—hematologist/other specialist N	4/49	9/38
Oncologic diagnosis suspected after first visit/not suspected N	15/38	15/32
Treatment ^3^ before correct diagnosis not administered/administered N	22/31	28/19
Number of specialists before correct diagnosis median (range)	3 (2–8)	3 (2–10)

^1^ CIRS—Cumulative Illness Rating Scale. ^2^ HADS—Hospital Anxiety and Depression Scale. ^3^ Either symptomatic treatment such as cough syrup for lymphoma, pain killers and massages for suspected radiculitis in myeloma or antibiotics for suspected infection.

**Table 2 medicina-55-00238-t002:** Factors included in B, C, D, E intervals (measured in log-transformed days) multiple linear regression models.

	Beta Value *	Percentile Beta Value **	t Value	*p* Value
**B ^1^ interval model (adjusted R^2^ = 0.052, *p* = 0.027)**				
(Intercept)	1.567	-	5.650	<0.001
CIRS value, range 3–12	−0.098	−9.3%	−2.367	0.019
HADS value, range 0-24	0.031	3.2%	1.501	0.136
**C ^2^ interval model (adjusted R^2^ = 0.332, *p* < 0.001)**				
(Intercept)	1.609	-	3.352	0.001
Number of specialists before correct diagnosis, range 2–10	0.252	28.7%	4.185	<0.001
Oncologic diagnosis suspected after 1^st^ visit: 1—Yes, 2—No	0.651	91.7%	3.139	0.002
Place of residence: 1—smaller cities, 2—Vilnius/Kaunas	0.343	40.9%	1.931	0.057
HADS value, range 0–24	0.032	3.3%	1.740	0.085
**D ^3^ interval model (adjusted R^2^ = 0.143, *p* < 0.001)**				
(Intercept)	2.626	-	9.901	<0.001
Diagnosis: 1—lymphoma, 2—MM	−0.605	−45.4%	−3.669	<0.001
Education: 1—no higher education, 2—higher education	−0.315	−27.0%	−1.932	0.056
Marital status: 1—married, 2—not married	0.265	30.3%	1.562	0.121
**E ^4^ interval model (adjusted R^2^ = 0.215, *p* < 0.001)**				
(Intercept)	3.002	-	7.344	<0.001
Place of residence: 1—smaller cities, 2—Vilnius/Kaunas	0.445	56.1%	2.727	0.008
HADS scale value, range 0–24	0.054	5.6%	3.124	0.002
Number of specialists before correct diagnosis, range 2–10	0.121	12.9%	2.331	0.022
Self-medication: 1—No, 2—Yes	0.436	54.7%	2.377	0.019

^1^—time between registration for first medical consultation and first medical consultation. ^2^—time between first medical consultation and diagnosis. ^3^—time between diagnosis and treatment. ^4^—overall time between symptom onset and diagnosis (sum of A, B, C intervals). * Beta values from log-transformed data. ** Percentile effect of independent variable each one-unit increase on untransformed independent variable.

**Table 3 medicina-55-00238-t003:** Longer diagnostic delay effect on risk of complications and Durie-Salmon stage—multiple binary and Poisson regression models.

	Beta Value	*p* Value	Odds Ratio	(95% CI)
**Factors included in MM ^1^ Complications (measured as ≤1 and >1) Regression Model**				
(Intercept)	−3.875	0.107	0.019	-
E ^2^ interval duration, range 23–1800 days	0.003	0.024	1.003	1.000–1.006
Gender: 1—male, 2—female	−0.752	0.255	0.471	0.129–1.720
Age, range 19–84 years	0.035	0.329	1.035	0.966–1.110
CIRS index value, range 3–12	−0.050	0.707	0.951	0.733–1.234
**Factors included in MM ^1^ Durie-Salmon stage (measured as 1 = 1A, 2 = IB, etc.) regression model**				
(Intercept)	0.644	0.218	1.904	-
E ^2^ interval duration, range 23–1800 days	0.001	0.049	1.001	1.000–1.001
Gender: 1—male, 2—female	−0.160	0.261	0.085	0.644–1.127
Age, range 19-84 years	0.010	0.216	1.010	0.994–1.026
CIRS index value, range 3–12	0.015	0.580	1.016	0.962–1.072

^1^—multiple myeloma. ^2^—overall time between symptom onset and diagnosis (sum of A, B, C intervals).

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
