# Peer review of "Time from Symptom Onset to Diagnosis and Treatment among Haematological Malignancies: Influencing Factors and Associated Negative Outcomes"

_medicina, 2019, doi:10.3390/medicina55060238_

Round 1
Reviewer 1 Report
This study investigates factors influencing diagnostic and or treatment delay in MM and lymphoma patients. The study identified a number of factors influencing diagnostic delay and suggest possible future interventions to reduce this time. Overall,the study addresses an important issue that can negatively affect overall outcome of patients with these malignancies.
Minor issues:
In Figure 1 there is a typo in the third box (the 't' is missing from first)
It would be worthwhile to define B symptoms and their relationship to lymphoma diagnosis either in the introduction or when introducing the term in the results.
The ISS for myeloma was revised in 2015 to give the R-ISS which encompasses the old ISS, chromosome abnormalities and serum LDH to give a better prognostic staging system. I would advise that the analysis in this study should be done using the R-ISS rather than ISS.
In Table 2 not all the information in the first column is visable
Author Response
Point 1: In Figure 1 there is a typo in the third box (the 't' is missing from first)
Thank you for the remark. This spell check mistake has been corrected.
Point 2: It would be worthwhile to define B symptoms and their relationship to lymphoma diagnosis either in the introduction or when introducing the term in the results.
We defined B symptoms in the results section.
Point 3: The ISS for myeloma was revised in 2015 to give the R-ISS which encompasses the old ISS, chromosome abnormalities and serum LDH to give a better prognostic staging system. I would advise that the analysis in this study should be done using the R-ISS rather than ISS.
Excellent remark and while we would certainly apply R-ISS rather than ISS in our future research work, at the time point this study started it was not possible. Only in 2018 R-ISS became standard of care at the time of diagnosis for all patients in our centre. We began interviewing patients in the beginning of 2017 and at that time they were already diagnosed and receiving treatment, meaning that diagnosis was established in approximately 2016-2017. In our defence, even recent studies regarding multiple myeloma patients are still using ISS and only now shifting to R-ISS, hopefully, making our work still clinically relevant.
Point 4: In Table 2 not all the information in the first column is visible
Thank you for the remark, corrected.

Reviewer 2 Report
Interesting topic, looking into an often neglected still very importan question, certainly worth publishing. Also, it is well written easy to read. Good references.
Only minor issues:
Table 1 could be devided All/MM/Lymphoma
In Table 1 last but one line Treatment before correct diagnosis – this requers some explanation does it mean steroids?
Line 141 We found statistically significant differences among A, B, C, D interval durations (p<0.001). I dont think this is a meaningful information we compare here completely different things.
Regarding delay between diagnosis and treatment. In younger patient IVF appointment can mean upto 14 days delay. Did the author collect this data?
Line 158 I would be more careful with the wording of thesse sentences suggesting that self medication and anxiety were actual factors casusing diagnostic delay. I think what the authors found is an association. And actually I would inheretnly think the other way around ie diagnostic delays can result in self medication and anxiety. Suggest to use ”associatied” insted of caused.
Table 2 is too complicated for the avarage reader (such as me), is there a possible graphic soulution for this? This is partially true about table 3 as well.
As there is such a few paper on this issue you might want to consider checking out: Varga, G., Mikala, G., Andrikovics, H., Masszi, T. How long does a myeloma patient currently wait for the diagnosis in Hungary?. Orv. Hetil., 2014, 155(39), 1538–1543
Author Response
Point 1: Table 1 could be divided All/MM/Lymphoma
Thank you for the remark. We divided it into lymphoma and myeloma sections.
Point 2: In Table 1 last but one line Treatment before correct diagnosis – this requires some explanation does it mean steroids?
Perfect remark, no it means either symptomatic treatment such as cough syrup for lymphoma, pain killers and massages for suspected radiculitis in myeloma or antibiotics for suspected infection. This explanation can now be found under our Table 1.
Point 3: Line 141 We found statistically significant differences among A, B, C, D interval durations (p<0.001). I dont think this is a meaningful information we compare here completely different things.
Thank you for the remark. In this case, we provided this information because we think it is important to show that differences between interval durations are significant and B and D intervals are the shortest. It helps to emphasize that attention needs to be drawn to shortening A and C interval durations, while B and D intervals seem to be acceptable already, which is one of our main conclusions. We are comparing different things (time intervals) but we are comparing them among the same individuals and then drawing conclusions from all 100 patients combined using different statistical tests, in our case Friedman test and Wilcoxon signed rank test. Both of these tests are designed to show whether there are differences between two (or more in case of Friedman test) related samples. Therefore, we believe it is statistically correct to compare how much time a patient spends in different time intervals and say that statistically significantly C interval does take the longest. In order to help our readers understand our logic behind this comparison, we explained it shortly in line 144-145, paragraph 3.2
Point 4: Regarding delay between diagnosis and treatment. In younger patient IVF appointment can mean up to 14 days delay. Did the author collect this data?
It is an excellent remark. We did ask every single patient what was the main reason for delay in each separate interval and then grouped their answers into categories which can be found in Supplement 6. Since answers are grouped we checked once more the original full answers and 2 youngest women actually waited around 2 weeks for IVF appointment. Our youngest male patient was 19 and he was diagnosed in 2016, while free of charge semen cryopreservation became available for male patients only in 2016 in Lithuania. Therefore, this patient possibly did not have a semen cryopreservation appointment. All of our MM patients were too old for IVF appointments. We corrected our Supplementary Figure 6 and included this reason for delay in D interval.
Point 5: Line 158 I would be more careful with the wording of thesse sentences suggesting that self medication and anxiety were actual factors casusing diagnostic delay. I think what the authors found is an association. And actually I would inheretnly think the other way around ie diagnostic delays can result in self medication and anxiety. Suggest to use ”associatied” insted of caused.
In regard to this remark we corrected mentioned sentence. We did avoid the word “caused” in all article and instead used gentler terms.
Point 6: Table 2 is too complicated for the avarage reader (such as me), is there a possible graphic soulution for this? This is partially true about table 3 as well.
We do agree that it is advanced statistics, however, we consulted our statistician and 4th author about various possible solutions. Since these tables represent regression models, standardized graphic solution is not available and tables are usually used for presenting regression analysis results. Therefore, we would like to leave our tables as it is, hoping that this would be useful statistical data for some readers.
Point 7: As there is such a few paper on this issue you might want to consider checking out: Varga, G., Mikala, G., Andrikovics, H., Masszi, T. How long does a myeloma patient currently wait for the diagnosis in Hungary?. Orv. Hetil., 2014, 155(39), 1538–1543
Thank you very much for this interesting article. We tried to find as many articles as possible, but data especially from European countries are lacking. We did include it in our discussion in 4 places. It is fascinating how completely different research groups found quite similar conclusions and suggest possible solutions in shortening diagnostic delay.
